# Does Smoking Affect OSA? What about Smoking Cessation?

**DOI:** 10.3390/jcm11175164

**Published:** 2022-08-31

**Authors:** Athanasia Pataka, Seraphim Kotoulas, George Kalamaras, Asterios Tzinas, Ioanna Grigoriou, Nectaria Kasnaki, Paraskevi Argyropoulou

**Affiliations:** Respiratory Failure Unit, G. Papanikolaou Hospital, Aristotle University of Thessaloniki, 54124 Thessaloniki, Greece

**Keywords:** obstructive sleep apnea, smoking, smoking cessation, nicotine replacement therapy, bupropion, varenicline, nortriptyline, clonidine, cytisine, treatment

## Abstract

The connection between smoking and Obstructive sleep apnea (OSA) is not yet clear. There are studies that have confirmed the effect of smoking on sleep disordered breathing, whereas others did not. Nicotine affects sleep, as smokers have prolonged total sleep and REM latency, reduced sleep efficiency, total sleep time, and slow wave sleep. Smoking cessation has been related with impaired sleep. The health consequences of cigarette smoking are well documented, but the effect of smoking cessation on OSA has not been extensively studied. Smoking cessation should improve OSA as upper airway oedema may reduce, but there is limited data to support this hypothesis. The impact of smoking cessation pharmacotherapy on OSA has been studied, especially for nicotine replacement therapy (NRT). However, there are limited data on other smoking cessation medications as bupropion, varenicline, nortriptyline, clonidine, and cytisine. The aim of this review was to explore the current evidence on the association between smoking and OSA, to evaluate if smoking cessation affects OSA, and to investigate the possible effects of different pharmacologic strategies offered for smoking cessation on OSA.

## 1. Introduction

Obstructive sleep apnea (OSA) is the most common sleep breathing disorder characterized by recurrent episodes of complete or partial obstruction of the upper airway, resulting in intermittent hypoxia and sleep fragmentation [1]. The main predisposing risk factors of OSA are male gender, older age, genetic and anatomical factors, central obesity, alcohol consumption, and also a narrowed upper airway. OSA is an independent risk factor for cardiovascular and cerebrovascular disease leading to increased morbidity and all-cause mortality [2]. 

Sleep fragmentation from frequent arousals result to daytime sleepiness, memory loss, and impaired cognitive function contributing to accidents [3]. OSA causes catecholamine surges, low-grade inflammation, and oxidative stress, having as a result cardio-metabolic consequences [4] such as hypertension, stroke, cardiac arrhythmias, and death when left untreated. The gold standard for the diagnosis of OSA is in laboratory polysomnography (PSG). As PSG is not widely available and rather expensive, other alternative diagnostic approaches have been tested as questionnaires, clinical prediction scores, and portable sleep monitors. Several questionnaires have been developed as screening tools for the detection of patients with a high probability of OSA, but none of these instruments achieved the reliability of PSG. Home sleep studies provide benefits in terms of cost and time, but with a lower diagnostic reliability compared with in-laboratory testing. However, in patients with a high probability of OSA and without co-morbidities, home sleep apnea testing is not inferior to in-laboratory PSG. Imagine modalities as lateral cephalometry, endoscopy, computed tomography (CT), or magnetic resonance imaging (MRI) may be useful for identifying the site of upper airway obstruction [4]. Continuous positive airway pressure (CPAP) is often considered the gold standard of OSA treatment; however, adherence to this treatment is often poor [5]. Additional OSA treatments include upper airway surgery, mandibular device therapy, modulation of hypoglossal nerve, pharmacotherapy, and combination therapy [6].

OSA is a heterogeneous disease with different clinical phenotypes and also different underlying pathogenetic mechanisms (endotypes). Several factors may be involved in the pathogenesis of OSA as (a) anatomical compromise resulting in a small, collapsible upper airway, (b) impaired pharyngeal dilator muscle function, (c) unstable oversensitive ventilatory control (elevated loop gain), and (d) low arousal threshold (individuals predisposed to wake up easily with respiratory disturbances). Additionally, the redistribution of body fluid may also be important [7,8,9,10]. Causes of narrowed upper airway anatomy that predispose to the development of OSA include nasal congestion, increased tonsil or adenoid or tongue size, fat deposition around the upper airway due to central obesity, and upper airway edema due to several factors including smoking [11,12,13,14,15,16,17]. The data on the relationship between OSA and smoking still remain controversial, failing to establish a safe conclusion about the association between these two entities [10].

Cigarette smoking is still the leading cause of preventable mortality and is the major contributor of cardiovascular disease. Both smoking and OSA increase the risk of cardiovascular disease as they both induce oxidative stress, endothelial dysfunction, and increase inflammatory response. For that they are associated with significant morbidity and mortality [16]. Further, there is evidence that each of these two conditions adversely affects the other and may lead to increased co-morbidity [15]. The health consequences of cigarette smoking are well documented; however, the effect of smoking cessation on sleep apnea has not been extensively studied. 

Targeting anatomy with medical therapy is challenging. Smoking cessation could improve OSA, as it may alleviate upper airways oedema, but studies to support this are limited [15,18]. Additionally, there is some evidence that links untreated OSA with increased smoking addiction [19], implying that the treatment of OSA could help towards a more successful smoking cessation attempt. The aim of this review was to investigate the current evidence on the association between smoking and OSA, to evaluate if smoking cessation may affect OSA and also to assess the possible effects of different pharmacologic strategies offered for smoking cessation on OSA.

## 2. Mechanisms by Which Smoking Can Result in OSA

Smoking induces chronic inflammation of the upper airway contributing to OSA symptoms [18,20]. Apart from active, passive and former smoking has been related with snoring [12]. There are studies that have confirmed the effect of smoking on sleep disordered breathing [11,12,13,14,15,16,17]. They have found that the prevalence of respiratory events, as sleep apnea or snoring, is higher in smokers, even in passive smokers [11,12,13,14,15,16,17]. Smoking has an impact on OSA through several mechanisms that include enhanced upper airway inflammation, changes of sleep architecture, instability of arousal mechanisms, and alterations of upper airway neuromuscular function [15] (Figure 1).

### 2.1. Upper Airway Inflammation due to Inhalation of Cigarette Smoke

Exposure to cigarette smoke may contribute to the inflammation of upper airways resulting in further narrowing and predisposing to the development of hypopneas and apneas. OSA alone has been found to be associated with inflammation of the upper airway [18]. In the study of Kim et al. [18], all the patients with moderate to severe OSA presented increased thickness and edema of the uvular mucosa in upper airway mucosal biopsy. Additionally, increased calcitonin gene-related peptide (CGRP) was found in the biopsies of smokers with OSA. This may have possibly further contributed to the inflammation of the upper airways of these patients. However further studies are needed to clarify the relationship of the inflammation caused by smoking and OSA pathogenesis.

Cigarette smoke has been found to reversibly activate hypoxia-inducible factor 1 (HIF-1α) [21] and this factor has been also found unregulated in OSA patients. As HIF-1α has been involved in the regulation of metabolic processes and in the development of insulin resistance and diabetes, it may further contribute to the development and aggravation of the metabolic co-morbidities of OSA [22].

### 2.2. Changes in Sleep Architecture and Smoking

Another mechanism by which smoking may have an impact on OSA pathogenesis is by its effect on sleep architecture [23,24,25]. There is evidence for an association between smoking and a worse sleep quality. Current smokers reported daytime sleepiness, difficulty falling and staying asleep, and increased insomnia symptoms [26] more frequently than non-smokers, with male smokers more likely to report nightmares [24]. Data from the Sleep Heart Health Study [27] demonstrated that current smokers, as compared with nonsmokers or former smokers, had longer sleep latency and a higher proportion of lighter sleep. However, no differences in sleep architecture were found between never and former smokers [27]. Analysis using EEG spectral analysis, demonstrated that the differences in sleep quality between smokers and nonsmokers were more evident during the early part of the night and decreased during the end [28]. The predominance of disturbances during the earlier part of the night may be attributed to the immediate effects of nicotine and/or to the nicotine withdrawal effects. Shorter sleep time, longer sleep latency, and higher rapid eye movement (REM) sleep density were found in smokers compared to non-smokers when matched by sex and age, in a study that used in-laboratory polysomnography [29]. Further, a larger analysis from 1492 adults that also used data from in-laboratory sleep studies, reported that current smokers had a higher arousal index and more sleep time with saturation of oxygen (SaO_2_) less than 90% compared with nonsmokers [30]. However, there is still a need for further objective data in order to support the hypothesis that current smoking is associated with poor sleep quality leading to OSA. 

### 2.3. Smoking May Affect Arousal Threshold

Arousals have been implicated in the pathogenesis of OSA [7,8,31]. There is a hypothesis that smoking results in a higher arousal threshold mostly from pediatric clinical studies, but there is still conflicting evidence. Due to nicotine’s short half-life, its effects on the arousal threshold may be different during the early period of the night compared with the end. In the study of Conway et al. [30], smokers presented a higher arousal index, with longer respiratory events and greater desaturations. On the other hand, infants that were exposed to maternal smoking demonstrated reduced arousability from sleep and passive smoking is a recognized risk factor for the development of sudden infant death syndrome [32,33]. Further data are needed as the higher arousal threshold may result in less instability of sleep and decrease the collapsibility of upper airways [23]. 

### 2.4. Smoking May Impair Upper Airway Neuromuscular Reflexes

The impairment of the protective upper airway neuromuscular reflexes by nicotine is considered to be another potential mechanism by which smoking may affect sleep apnea. Data from animal models have shown that exposure to passive smoking in lambs resulted in enhanced respiratory inhibition and in more apneas with laryngeal stimulation [34]. Studies in animal models support the hypothesis that nicotine exposure may enhance the constriction of the upper airways; however, data in humans to support this are scarce.

## 3. Association between Smoking and OSA 

The association between smoking and OSA is not currently well-established (Figure 1). Active but not former smoking was associated with a higher possibility of developing moderate or severe OSA in the Wisconsin Sleep Cohort Study, even after adjusting for confounding factors, and especially in heavy smokers [14]. Other smaller studies showed that OSA and smoking were independently associated, and that the prevalence of current smoking was higher in OSA patients compared with those that did not suffer from OSA [35,36]. Similarly, it was found that smoking related with earlier age of OSA diagnosis and that heavy smokers suffered from more severe disease [37]. Likewise, in a large single-center study that included 3613 OSA patients, it was reported that smokers suffering from OSA presented higher AHI and lower mean oxygenation during sleep [38]. A more recent study had also demonstrated the significant effect of increased smoking status, expressed by Pack/Years (P/Ys), with OSA severity, expressed by Apnea Hypopnea Index (AHI) and oxygen desaturation index (ODI) [39]. Our group had recently performed a study evaluating the effects of smoking on OSA and even if we did not find an independent effect of smoking on OSA, the severity of smoking status (measured by number of cigarettes/days, P/Ys), and nicotine dependence were found higher in patients with more severe OSA. Smoking was not significantly associated with OSA after adjusting for gender, age, BMI, and alcohol [40]. Similarly, another study reported that smoking was not associated with AHI but only with ODI and arousals, with these events been more pronounced in current smokers [30]. Those that smoked more than 15 P/Ys presented higher arousal index and longer time spend with SaO_2_ < 90%. Former smokers with a history of more than 15 P/Ys presented higher arousal index and AHI compared with those with less P/Ys [30].

On the other hand, there are other studies that did not find a causal relationship between cigarette smoking and OSA. Older studies have demonstrated that smokers present a significant decrease in nocturnal oxygen saturation, but when compared with non-smokers, no significant differences in either AHI or ODI were found [41]. However, evidence from the Sleep Heart Health Study showed that the former, but not current smoking was associated with more severe disease. An inverse association between current smoking and AHI was observed, with lower AHI values in current smokers [42]. Additionally, a large population study had also found that current smoking was strongly but inversely related with self-reported OSA both in men and women [43]. A large cross-sectional study reported that, after adjusting for age, BMI, and gender, smoking was not an independent risk factor for OSA. However, the same study found that patients with more severe disease (AHI > 50) were heavier smokers and also that compared with non-smokers, heavy smokers presented higher AHI [44]. Similarly, another study also concluded that although there was no significant association between smoking and OSA, smokers presented higher AHI [45]. In a retrospective analysis, when current/former smokers were compared with non-smokers, no significant differences in the AHI were reported but a lower nocturnal mean oxygen saturation was found in current/former smokers [46]. These three studies [44,45,46] reported that smokers presented worse daytime sleepiness that was attributed to nicotine’s effects on the upper airway, on sleep architecture, and also to nocturnal hypoxia. In a more recent meta-analysis, there was evidence to confirm that OSA was related with alcohol use, but not with tobacco or caffeine. However, the level of evidence in that analysis was low and the results should be interpreted with caution [47]. 

The differences between the results of the aforementioned studies could be explained by the different populations that were examined, as some of the studies evaluated populations from the community setting and others population referred to a sleep clinic. Additionally, according to CDC (https://www.cdc.gov/nchs/nhis/tobacco/tobacco_glossary.htm, accessed 2 August 2022) a current smoker is defined as an adult who has smoked 100 cigarettes in his or her lifetime and who currently smokes cigarettes, and a former smoker as an adult who has smoked at least 100 cigarettes in his or her lifetime but who had quit smoking at the time of the study. Passive smoking or second-hand smoke is the inhalation of environmental tobacco smoke. A heavy smoker is the smoker who consumes 20 or more cigarettes per day. (https://www.canada.ca/en/health-canada/services/health-concerns/tobacco/research/tobacco-use-statistics/terminology.html, accessed 2 August 2022) [37]. However, it is worth mentioning that in some studies the aforementioned criteria may differ, i.e., a heavy smoker defined as someone who smokes more than 40 cigarettes per day [39], or a current smoker someone who smokes more than 10 pack/years [41] and this may also explain the differences in the results.

## 4. Does OSA Predispose to Smoking?

There are very limited data evaluating the question whether OSA predisposes to smoking. There is a hypothesis that individuals with undiagnosed or untreated OSA use the stimulating effects of nicotine by smoking cigarettes, in order to cope with their daytime symptoms such as sleepiness or loss of concentration. Societal demands, such as mental alertness or even attempts for weight loss, may play their role in tobacco use in sleepy obese patients with OSA [14,19]. Nicotine has the property to increase the levels of dopamine in the nucleus accumbens. This induces the feeling of reward and arousal, which make the individual addicted to nicotine, especially if chronically depressed or sleepy [46].

Depression is rather common in patients with OSA [4] and this can also be a contributing factor for smoking addiction in these patients. Nicotine stimulates the serotonergic neurons in the dorsal raphe nucleus of the pons and this stimulation has been associated with the mood improvement of patients with major depression that smoke [48]. It has been found that the administration of nicotine subcutaneously alters the discharge rate of dorsal raphe serotonergic neurons during REM sleep [49]. On the other hand, chronic hypoxia may be associated with dopamine release in the carotid body augmenting ventilation and encouraging nicotine addiction by stimulating the reward center [50].

It has been also hypothesized that long-term hypoxia (as in OSA) through adaptation may be related with the increase in the number of the binding sites of nicotine observed in smokers such that it further contributed to the vicious cycle of smoking addiction. The more nicotine receptors available, the greater the increase of smoking frequency [51]. Sleep fragmentation and sleep loss, which are the main consequences of OSA, were found to enhance pain sensitivity, especially through inflammatory and hypoxia markers, as HIF 1a [52]. Smoking has been related to dysregulated pain processing and the development of persistent pain. On the other hand, pain has been reported to be a barrier to smoking cessation, as smokers with pain presented more severe withdrawal symptoms and experienced more difficulties during quitting attempts with more frequent relapses [53].

## 5. Nicotine and Sleep

Nicotine stimulates a7 and a4b2 nicotinergic acetylcholine (nAChR) receptors and alters indirectly the serotonergic, dopaminergic, and glutaminergic systems in the brain [54]. Smoking results in physical and psychological dependence. The most common withdrawal symptoms include cravings for tobacco, loss of concentration, irritability, restlessness, mood changes, depression, constipation, increased appetite, and sleep disturbances [55]. The most prevalent sleep disturbance reported is insomnia [55]. Smoking has been associated with the development of sleep disturbances in population studies with large numbers of subjects [14]. However, the results of different studies are not comparable as smoking history, degree of nicotine dependence, and smoking cessation time varied between them [56]. Depression, the use of alcohol and caffeine, co-morbidities, and the use of different medications were not assessed in all studies. Smokers often have additional unhealthy lifestyle habits, as the use of caffeine and alcohol can also enhance sleep disorders [56]. Data from animal studies demonstrated that nicotine had an arousing effect, enhancing wake time and decreasing slow wave sleep (SWS). Additionally, a dose-dependent effect of nicotine on REM sleep was found, with lower doses having stimulating effects, higher doses suppressing effects on REM sleep, with rebound effects during nicotine discontinuation [56].

Studies in humans that investigated the effect of nicotine consumption and sleep with subjective measurements showed that smokers have almost double the risk of sleep disturbances, especially with difficulties falling asleep, resulting in daytime sleepiness [14,24]. Apart from sleep disturbances, other studies also noted the association between smoking, alcohol and coffee consumption, and additionally anxiety and depressive symptoms [25,57]. Chronic insomnia and changes on sleep architecture differed according to smoking intensity, with smokers with heavy and regular cigarette consumption developing some tolerance to nicotine’s effects on sleep [57].

Studies on humans investigating the effects of nicotine on sleep using objective measurements by polysomnography (PSG), found prolonged sleep latency and REM latency, reduced sleep efficiency, total sleep time (TST) and SWS, but no effect on arousals, in smokers compared with non-smokers [27,58]. However other studies reported an increased number of arousals and nocturnal awakenings with increased frequency of the changes of sleep stages but with unchanged TST [59]. A study that used spectral analysis found decreased d-frequencies and increased a-frequencies in the electroencephalographic activity of smokers that were more pronounced in the beginning of the sleep [28].

## 6. Smoking Cessation and Sleep

During nicotine withdrawal, subjective impairments of sleep have been reported with the severity of sleep problems having been associated with the duration of abstinence and the degree of nicotine dependence. Nicotine withdrawal symptoms usually begin 3–12 h after smoking cessation, reaching to the maximum intensity in 1–3 days, and they may last up to 3 or more weeks [55]. Of most of the studies examined, only the first three or four nights after withdrawal covered only the initial part of abstinence symptoms. A study that investigated withdrawal symptoms for a period of 21 days found that there was an increased frequency of sleep problems, especially insomnia symptoms, during withdrawal that may have increased the relapse rate of smoking cessation [60].

Smokers often report insomnia symptoms [15], increased arousals, longer N1 and N2 sleep stages, and shorter SWS [58,61]. Alternatively, during smoking cessation, smokers also report insomnia, shorter sleep duration and sleep efficiency, and longer sleep latency with frequent awakenings, especially during the first days to first weeks [15]. Additionally, PSG studies performed during the week following nicotine withdrawal in heavy cigarette smokers have shown increased sleep fragmentation and increased number of awakenings after sleep onset [58,59]. Subjective sleep was found reduced by about 30 min and the number, as well as the duration, of awakening increased, with a maximal effect around the second withdrawal day [58].

Disturbed sleep during the smoking cessation process might worsen the abstinence rates and may be a risk factor for relapse [61,62,63]. For that there is a need for smoking cessation therapies to adapt to the specific requirements of each smoker. Smokers with concomitant depression may smoke more and may experience more depressive episodes during withdrawal that may negatively affect their smoking cessation attempt. Additionally, these patients often suffer from sleep disorders, especially insomnia and changes in REM sleep [64].

## 7. Smoking Cessation Treatments and OSA

Tobacco dependence is a chronic disorder with a high risk of relapse after tobacco cessation. The key components of successful cessation are the combination of education, behavioral support, and pharamacotherapy of the smoker. Nicotine Replacement Therapy (NRT), bupropion, and varenicline are considered as first line medications for smoking cessation, whereas nortriptyline and clonidine are second line. Other medications, such as cytisine, have also been used for smoking cessation [65].

### 7.1. Nicotine Replacement Therapy

Nicotine replacement therapy (NRT) is used to replace the nicotine of cigarettes but without the harmful effects of smoke. NRTs reduce withdrawal symptoms and allow smokers to focus on changing their behavior towards smoking and quit [65]. The efficacy of nicotine replacement therapies (NRTs) to relieve craving for cigarette has been recognized but the studies on NRT capacity to improve tobacco withdrawal sleep disturbances are less and controversial. The transdermal nicotine patch is the most commonly purchased NRT, and it delivers nicotine to the blood at a slow constant rate through topical application on the skin. There are studies indicating that NRTs increase sleep disruptions when abstinent smokers were assessed with self-report questionnaires. When the effects of NRTs were studied objectively with PSG recordings, NRTs seemed to improve the objective signs of sleep disturbances but not the subjective symptoms [66]. The 24-h NRTs appeared more effective than a 16-h NRTs in improving objective indicators of sleep quality by increasing SWS and lowering microarousals of smokers undergoing smoking cessation [67]. NRTs affect the sleep of non-smokers and former smokers by the same manner. It has been found that the administration of transdermal nicotine patch in non-smokers resulted in earlier wake-up times, increased sleep stage 2, and REM sleep dose-dependent reduction [68]. During the nights following the cessation of the nicotine patch, REM sleep rebound was found, with unchanged sleep latency, sleep continuity, and TST [68].

#### 7.1.1. The Effects of NRT on Sleep of Smokers during Smoking Cessation

Studies that investigated the effect of the nicotine patch during withdrawal using subjective sleep variables found that those who used nicotine patches reported more sleep difficulties, but additional smoking was not evaluated in all studies [69,70]; with the effects of smoking and that of patches contributing to higher nicotine levels. On the other hand, other studies noted that sleep problems persisted with the use of nicotine patches in a controlled smoking cessation program but occurred more frequently in the placebo group suggesting nicotine’s withdrawal effects [15,61,70]. Additionally in another study, sleep problems were reported especially during the first week of nicotine replacement therapy, that then decreased with use of low-dose nicotine patches and by continuing abstinence [71]. The duration of nicotine abstinence and higher nicotine dependency were predictors for side effects, including sleep disturbances [70]. As both nicotine administration and withdrawal may similarly affect sleep, it is difficult to be distinguished. A positive correlation was found between sleep problems and the level of cotinine in plasma with the participants with sleep disorders having higher levels of cotinine while using NRTs [72]. 

Studies that investigated the effect of nicotine patches on the sleep of smokers during smoking cessation with PSG demonstrated an increased number of arousals but with no alterations in sleep architecture in smokers under placebo [66]. Using nicotine patches with ongoing abstinence resulted in decreased arousals and stage 2 sleep, as well as in increased SWS [66]. However, all participants (those using NRT and those on placebo) reported reduced sleep quality and frequent awakenings during withdrawal. The time of NRTs application is important, as nicotine substitution with patches in the morning provides the smoker steady nicotine supply during the day and their removal before bedtime mimics the smoker’s behavior with abstinence during night. The fall of nicotine levels during the night may result in symptoms of withdrawal, especially in heavy smokers that wake up in order to smoke during the night, implying high nicotine dependence [73]. When nicotine patches were applied only at night, the number of arousals and total wake time increased, REM sleep decreased, and dreams were more intensive compared with placebo [74]. 

The time of application of the patch is important as the maximum nicotine’s plasma concentrations are reached after 8–10 h [75]. When 24 h nicotine patches were compared with 16 h, it was found that the 24 h patches led to lower Profile of Mood Scores (POMS) and craving the next morning with no differences in the subjective sleep [76]. Both 24 h and 16 h patches application, resulted in shorter total sleep time, prolonged sleep latency [67], whereas NREM sleep, SWS, and arousals were improved with the 24 h patch. Additionally, during the first third of the night, REM density increased and also with the 24 h patch, an increase of b-activity during REM sleep was demonstrated, whereas a reduction was found with 16 h patch [67]. 

It is important to mention that the studies evaluating the effect of NRTs on sleep during smoking cessation were not comparable as they had different methodology i.e., used different nicotine dose, different time of NRTs’ application, some allowed additional cigarette smoking, and most of the studies did not individualize the dose of NRTs according to the smokers’ dependence. The variability in nicotine metabolism may lead to withdrawal symptoms at different times in each smoker that may implicate nicotine’s effects on sleep. There was a discrepancy between the data provided from subjective and objective sleep evaluation. Using subjective parameters, disturbed sleep was reported during smoking cessation both with and without the use of NRTs. On the contrary, studies using objective measures, as PSG, noticed a positive effect of NRTs [67]. In order to achieve the best results, it was important to adapt an individual effective dose of NRTs, as sleep impairment had been closely linked with withdrawal symptoms, such as craving [56].

#### 7.1.2. Effects of NRT for OSA Treatment

Nicotine has stimulant effects on ventilatory drive and genioglossus muscle [56]. It seems that the effects of cigarette smoke and nicotine on the upper airways might differ. There are studies that have evaluated the hypothesis that nicotine could be helpful for the treatment of OSA with conflicting results. A study that used nicotine gum in a total dose of 14 mg in eight patients with OSA and different smoking habits, with two presenting hypoventilations, showed a reduction in the number of obstructive and mixed apneas in the first 2 h of sleep when nicotine dilating properties on the upper airway were still effective. However, due to nicotine’s short half-life (2–4 h), as the levels in the blood declined, the upper airway resistance and AHI increased during the night. No effect of nicotine was observed on central apneas, sleep architecture, and end-tidal CO2 during wakefulness [77].

However, other studies that administered nicotine via patches [78] or via a tooth patch did not find any improvement in respiratory events [79]. Non-smoking patients with OSA were randomized to receive 11 mg of transdermal nicotine patch for 12 h, or a placebo, with no improvements in snoring or respiratory events [78]. However, a negative correlation was found between serum nicotine concentration and the mean duration of apneas and hypopneas. Sleep efficiency and TST decreased with the administration of nicotine patch. Nicotine tooth patches were used in another study in doses of 2 and 4 mg in patients with OSA [79]. Nicotine tooth patches release nicotine continuously with high nicotine levels measured in saliva for almost 4 h but in low levels in the plasma. No improvement of AHI or sleep stages was found in this study. Overall, these small studies suggest that NRTs did not improve the respiratory events of OSA patients.

### 7.2. Bupropion

Bupropion is an antidepressant drug prescribed for disorders such as major depressive disorder, attention-deficit/hyperactivity disorder, and seasonal affective disorder [80]. Nicotine has anti-depressive effects, and it has been postulated that chronic smokers try to control their depressive symptoms by using nicotine. Bupropion sustained release (Zyban^®^) is a non-nicotine oral therapy used for smoking cessation. Bupropion reduces withdrawal symptoms associated with smoking cessation. It is an inhibitor of the neuronal uptake of dopamine and noradrenaline with minimal effects on serotonin. The mechanism by which bupropion enhances smoking abstinence is not fully known; it is assumed that the smoking cessation action is mediated by dopaminergic and/or noradrenergic mechanisms [81]. The advantages of bupropion include a relatively low risk of inducing restless leg syndrome [82] or REM sleep behavioral disorder [83,84]. On the other hand, its alerting effects may lead to sleep disruption, particularly in the short-acting formulations administered late during the day [85].

Despite bupropion being used for many years for smoking cessation, there is a paucity of data concerning its effects on OSA. In studies evaluating the effects of bupropion as an antidepressant, it has been found that it has negative effect on sleep continuity increasing insomnia symptoms [81]. Bupropion, unlike other antidepressants, does not have REM-suppressant effects and on the contrary, may decrease REM latency and increase REM [86]. In OSA patients that present increased respiratory events during REM, this effect may worsen their disease severity.

### 7.3. Varenicline

Varenicline (Champix^®^) is one of the most effective medications used towards smoking cessation, as it significantly increases the smoking abstinence rates [87]. It affects the nervous system as it binds with a high affinity and selectivity with alpha4beta2 nicotinic acetylcholine receptors (α4β2nAChRs), where it acts as a partial agonist, with lower efficacy than nicotine, reducing craving for cigarettes, but also acts as an antagonist reducing the pleasurable, and reinforcing the rewarding effects of smoking in the presence of nicotine. Additionally, it stimulates dopamine release by its acetylcholinergic action [88]. Varenicline has been used as an aid for smoking cessation for almost 15 years and its most common side effects include nausea, headaches, insomnia, and abnormal dreams. Studies have proven that there is no evidence of an increased neuropsychiatric risk (i.e., suicidal ideation, suicide or attempted suicide, depression) in smokers receiving varenicline [89]. However, sleep disorders such as insomnia and abnormal dreams are also a common symptom during smoking cessation. Insomnia-related symptoms of varenicline peak in the first week of a quitting attempt and decline progressively at 2–12 weeks [90]. There is evidence that a high percentage of smokers treated with varenicline report sleep problems, that dissolve across time [61]. Apart from insomnia and disturbed sleep, other side effects of varenicline are abnormal dreams, nightmares, and rarely somnambulism and REM sleep disorders [91,92].

In a study based on sleep diaries, no significant changes were found in sleep measures after varenicline administration, but frequent awakenings and abnormal dreams were documented [92]. Alternatively, current smokers often complained of fragmented sleep and decreased sleep time, mostly during the second part of the night, probably due to the tobacco withdrawal state occurring each night [93]. Restless Leg Syndrome amelioration was also reported with the administration of varenicline [94] and this had been attributed to dopamine release stimulation from varenicline. In a recent study, we have shown that varenicline prolonged sleep latency, the latency of N2 and N3 stages, and reduced the AHI, especially during REM sleep in smokers suffering from OSA. The arousal index increased in OSA patients, but this did not have a significant effect on respiratory events. However, it was difficult to distinguish between the effects of smoking cessation per se or the adverse effects of the medication [95]. Further studies are needed to explore the mechanisms and applications of varenicline’s effect on sleep. Unfortunately, varenicline has been recalled from European market for almost a year due to the presence of N-nitroso-varenicline above acceptable intake limits [96].

### 7.4. Nortriptyline

Nortriptyline is a tricyclic antidepressant indicated for the treatment of depression (FDA-approved). However, it can also be used off-label for chronic pain, diabetic neuropathy, orofacial pain, postherpetic neuralgia, and as a second line medication for smoking cessation. Its potential side effects include nausea, dry mouth, constipation, headaches, sedation, and risk of arrhythmia in patients suffering from cardiovascular disease. Nortriptyline application for smoking cessation is rather limited due to its potential side effects. The effects of nortriptyline in patients with OSA have not been examined. In a study that evaluated the effects of nortriptyline on subjective and EEG sleep measures in elderly patients suffering from major depression over a year of treatment, showed that nortriptyline increased phasic REM activity, decreased REM sleep acutely and persistently, and possibly due to that, decreased sleep apnea with no effect in periodic limb movements during sleep [97].

Protriptyline, a tricyclic antidepressant closely related to nortriptyline, was believed to stimulate ventilation in patients suffering from sleep apnea. There is evidence from older studies that protriptyline decreased AHI, but this effect is controversial. No significant change in the duration and frequency of sleep-disordered breathing (SDB) during non-REM sleep was observed after treatment with protriptyline, but a decreased number of apneas and of the peak fall in oxygen saturation was found. In addition, a reduction in the amount of REM sleep was also observed resulting in a further reduction of more severe respiratory events during that time [98]. In another small double-blind crossover study, protriptyline improved daytime somnolence and nocturnal oxygenation, but with no significant changes on respiratory events. REM sleep was reduced resulting in decreased REM apneas that remained after six months of treatment [99]. We did not find similar studies evaluating nortriptyline effects on OSA.

### 7.5. Clonidine

Clonidine is an α2-adrenergic agonist used for the treatment of hypertension, with off label uses for attention deficit hyperactivity disorder, menopausal flushing, restless legs syndrome, certain pain conditions such as postoperative and dysmenorrhea, and drug withdrawal (alcohol, opioids, or nicotine) [100]. Clonidine is a second-line treatment for smoking cessation; however, such as nortriptyline, it is not registered for smoking cessation therapy worldwide [65]. The most common side effects of clonidine include postural hypotension, drowsiness, fatigue, and dry mouth, which limits its usefulness for smoking cessation.

The effect of clonidine hydrochloride administered for ten days was evaluated in eight men with OSA. REM sleep latency increased whereas REM sleep was suppressed resulting in the reduction of respiratory events during that time. However, clonidine had no effect on the respiratory events during non-REM [101]. We did not find other studies evaluating clonidine’s effects on OSA.

### 7.6. Cytisine

Cytisine is an alkaloid present in a number of plants that was used in eastern and central European countries as a respiratory stimulant and as an aid for smoking cessation since 1960. Until very recently, there was a renewed interest about cytisine for smoking cessation due to its low cost as compared with other smoking cessation medications [102]. The most common side effects of cytisine include nausea, vomiting, and sleep disorders [102]. Cytisine’s activity is very similar to that of nicotine, as it is a partial agonist of α4β2 and a near full agonist of α6β2 cholinergic nicotinic acetylcholine receptors (nAChRs). At low doses, cytisine activates nAChRs in the mesolimbic reward pathways and that inhibits nicotine’s sensory stimulation and limits the symptoms of withdrawal [103]. Cytisine was not only noninferior to NRT but had superior effectiveness [104]; however, in a recent study, varenicline was found more effective [105]. Varenicline is a partial α4β2 agonist that has all the characteristics of cytisine but is more efficient with fewer side effects [106]. We have not found any studies on the effects of cytisine on patients with OSA. 

A summary of the effects of medications used for smoking cessation on sleep and OSA are presented in Table 1.

## 8. Incidental Effects of Smoking Abstinence on OSA

Weight gain is a consequence of smoking cessation. The median weight gain after cessation is around 2–3 kg [107]. On the other hand, due to withdrawal symptoms, around 10% of smokers trying to quit experience a greater increase of weight, about 13–15 kg. Increased weight is a risk factor of OSA, and the excessive weight gain due to smoking cessation may aggravate the existing OSA [1].

Sleep disturbance in smokers may also be attributed to periodic limb movement disorder (PLMD) [108]. Smoking (>20 cigarettes/day) has been associated with restless legs syndrome (RLS) occurrence [109], where PLMs are highly prevalent. PLMs are related to impaired dopaminergic function and nicotine, as a dopamine agonist may influence PLM symptoms by the central dopamine system. Further studies are needed for the relationship between nicotine, smoking cessation, and PLMs, as well as the possible use of nicotine as NRTs for the treatment of PLMs.

## 9. Conclusions

The association between smoking and OSA is not currently well-established as there are studies that support this association and others that do not. Smoking is associated with insomnia, and smokers experience increased arousals, longer N1 and N2 sleep stages, and shorter SWS. OSA may be predisposed to smoking, as individuals with undiagnosed or untreated OSA smoke in order to use the stimulating effects of nicotine, especially those with depressive symptoms. Sleep disturbances may affect the success of smoking cessation attempts since sleep disruption may contribute to other symptoms as irritability or concentration difficulties. Sleep deprivation and daytime sleepiness may contribute to the abstinent smokers being less able to cope with everyday obligations making them more vulnerable to relapse.

The effect of smoking cessation in OSA is less clear. The effect of NRTs on OSA was evaluated only in small studies that did not show a significant improvement of the respiratory events. Varenicline use in smokers suffering from OSA lead to prolonged sleep latency, the latency of N2 and N3 stages and reduced the AHI, especially during REM sleep. The scope of this review was to investigate the relationship of smoking and especially smoking cessation with OSA. We did not find any studies about the association of e-cigarettes or smokeless tobacco and OSA. On the other hand, these products are not considered as safe strategies for smoking cessation. Further investigations on the relationships and possible implications for treating sleep disorders in smokers during smoking cessation are needed, as it is difficult to distinguish between the effects of smoking cessation or the adverse effects of the smoking cessation pharmacotherapy.

## Figures and Tables

**Figure 1 jcm-11-05164-f001:**
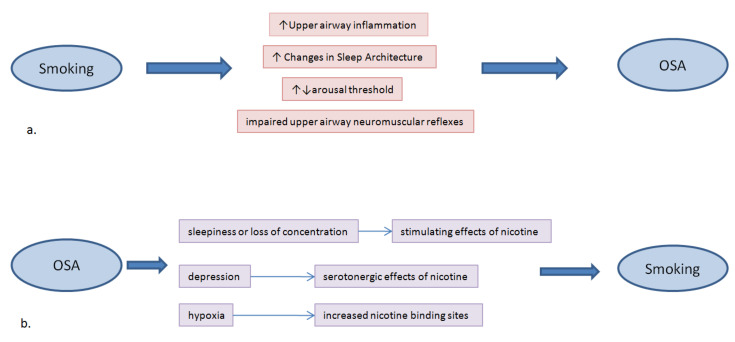
A summary of the bidirectional relationship between Obstructive Sleep Apnea (OSA) and cigarette smoking; (**a**) Mechanisms by which smoking can result in OSA; (**b**) Associations between OSA and smoking.

**Table 1 jcm-11-05164-t001:** Summary of the effects of medications used for smoking cessation on sleep and OSA.

NRT—smoking cessation	Subjective measures: deterioration of sleep quality—increase sleep disruptions and awakenings [66] PSG recordings: 24-h NRTs: improved sleep quality—shorter TST, prolonged sleep latency but increased SWS, decreased microarousals, increase of b-activity during REM [67] 16-h NRTs: shorter total sleep time, prolonged sleep latency, reduction of b-activity during REM [67] NRT patches applied at night: number of arousals and total wake time increased, REM sleep decreased and more intensive dreams [74] During the nights following the cessation of NRT: REM sleep rebound, unchanged sleep latency, sleep continuity and TST [68]
NRT—OSA treatment	nicotine gum: reduction of the number of obstructive and mixed apneas in the first 2 h of sleep but upper airway resistance and AHI increased during the night [77] transdermal or tooth patch: no improvement in respiratory events [78,79]
Bupropion	alerting effects increasing insomnia symptoms [85] does not have REM-suppressant effects but may decrease REM latency and increase REM [86] may increase OSA severity due to increased respiratory events during REM no specific studies on OSA patients
Varenicline	insomnia, disturbed sleep, frequent awakenings, abnormal dreams, nightmares, rarely somnambulism and REM sleep disorders [91,92] Restless Leg Syndrome amelioration [94] in smokers with OSA: reduced AHI, especially during REM, prolonged sleep latency, N2 and N3 latency, increased arousal index [95]
Nortriptyline	increased phasic REM activity, decreased REM sleep acutely and persistently, and possibly due to that, decreased sleep apnea [97] no effect in periodic limb movements during sleep [97]. no specific studies on OSA patients
Clonidine	REM sleep latency increased, decreased REM, reduction of respiratory events during REM no effect on the respiratory events during non-REM [101]
Cytisine	no specific studies on the effects on sleep or OSA

NRT = Nicotine replacement therapy, PSG = polysomnography, TST = total sleep time, SWS = slow wave sleep, REM = rapid eye movement, AHI = Apnea hypopnea index.

## Data Availability

Not applicable.

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
