# Peer review of "Does Smoking Affect OSA? What about Smoking Cessation?"

_jcm, 2022, doi:10.3390/jcm11175164_

Round 1

Reviewer 1 Report

This is a well-written and insightful review focusing on the effect of smoking and smoking cessation on obstructive sleep apnea. It adds to the literature and the authors have cited several relevant previous studies regarding the main subjects and key-points of the paper covering a wide field of the topic. 

I would like to highlight some notes you may consider adding to your manuscript.

1.     In my opinion in the introduction, you should include additional background information regarding the diagnostic methods of OSA (polysomnography, imaging modalities, questionnaires, scores, etc).

2.     Please add definitions of what you (or the mentioned studies) considered current smoking, former smoking, heavy smoking and passive smoking. It is worth mentioning that some studies may differ in the defined criteria for these subgroups. For example, some may consider current smoking as a patient who has been actively smoking for over a year or has smoked 100 cigarettes in their lifetime, etc.

3.     You may consider adding studies investigatin any connection between OSA and the usage of tobacco products aside from cigarettes, also smokeless tobacco products if available.

L 134: “passive smoking in lamps” is misspelled, should be “lambs”.

L 699: Citation 98 is missing.

Author Response

We would like to thank the reviewer for all the comments that helped us to improve our manuscript.

 This is a well-written and insightful review focusing on the effect of smoking and smoking cessation on obstructive sleep apnea. It adds to the literature and the authors have cited several relevant previous studies regarding the main subjects and key-points of the paper covering a wide field of the topic. 

I would like to highlight some notes you may consider adding to your manuscript.

  • In my opinion in the introduction, you should include additional background information regarding the diagnostic methods of OSA (polysomnography, imaging modalities, questionnaires, scores, etc).

Answer:

Thank you for your comment we have added some information regarding the diagnosis of OSA.

‘The gold standard for the diagnosis of OSA is in laboratory polysomnography(PSG). As PSG is not widely available and rather expensive, other alternative diagnostic approaches have been tested as questionnaires, clinical prediction scores and portable sleep monitors. Several questionnaires have been developed as screening tools for the detection of patients with high probability of OSA, but none of these instruments achieved the reliability of PSG. Home sleep studies provide benefits in terms of cost and time, but with lower diagnostic reliability compared with in-laboratory testing. However, in patients with a high probability of OSA and without co-morbidities, home sleep apnea testing is not inferior to in-laboratory PSG. Imagine modalities as lateral cephalometry, endoscopy, computed tomography (CT) or magnetic resonance imaging (MRI) may be useful for identifying the site of upper airway obstruction.’

  • Please add definitions of what you (or the mentioned studies) considered current smoking, former smoking, heavy smoking and passive smoking. It is worth mentioning that some studies may differ in the defined criteria for these subgroups. For example, some may consider current smoking as a patient who has been actively smoking for over a year or has smoked 100 cigarettes in their lifetime, etc.

Answer:

Thank you for your comment we have added some information regarding the definition of current smoking, former smoking, heavy smoking and passive smoking.

According to CDC (https://www.cdc.gov/nchs/nhis/tobacco/tobacco_glossary.htm last assessed 02/08/22) Current smoker is an adult who has smoked 100 cigarettes in his or her lifetime and who currently smokes cigarettes. Former smoker is an adult who has smoked at least 100 cigarettes in his or her lifetime but who had quit smoking at the time of the study. Inhaling Environmental Tobacco Smoke (ETS) is defined as passive smoking or second-hand smoke. Heavy smoker is a smoker who consumes 20 or more cigarettes per day. (https://www.canada.ca/en/health-canada/services/health-concerns/tobacco/research/tobacco-use-statistics/terminology.html  last assessed 02/08/22). However, it is worth mentioning that in some studies the aforementioned defined criteria may differ, i.e heavy smoker defined as someone who smokes more than 40 cigarettes per day[39], or current smoker someone who smokes more than 10 pack/years [41] and this may also explain the differences in the results.

  • You may consider adding studies investigating any connection between OSA and the usage of tobacco products aside from cigarettes, also smokeless tobacco products if available.

Answer:

The scope of this review was to investigate the relationship of smoking and especially smoking cessation with OSA. We did not find any studies about the association of e-cigarettes or smokeless tobacco and OSA. On the other hand these products are not considered safe strategies for smoking cessation. There are studies on the effect of e cigarettes on sleep ( for example Brett EI, Miller MB, Leavens ELS, Lopez SV, Wagener TL, Leffingwell TR. Electronic cigarette use and sleep health in young adults. J Sleep Res. 2020 Jun;29(3):e12902. doi: 10.1111/jsr.12902. Epub 2019 Sep 4. PMID: 31486154; PMCID: PMC7299171., So CJ, Meers JM, Alfano CA, Garey L, Zvolensky MJ. Main and Interactive Effects of Nicotine Product Type on Sleep Health Among Dual Combustible and E-Cigarette Users. Am J Addict. 2021 Mar;30(2):147-155. doi: 10.1111/ajad.13130. Epub 2020 Nov 24. PMID: 33231910.) indicating worse sleep health among e cigarette or dual users. However these studies are not within the subject of our review.

We have added this comment in the conclusions of the revised manuscript.

  • L 134: “passive smoking in lamps” is misspelled, should be “lambs”.

Answer:

Thank you for your comment we have corrected the mistake

  • L 699: Citation 98 is missing.

Answer:

Thank you for your comment we have corrected the mistake

Reviewer 2 Report

The undertaken manuscript widely describes the relationship between cigarette smoking and OSA, which is an important topic in clinical practice, possibly with too little focus in the present research. 

Several notes:

-       While describing possible mechanisms connecting the OSA and cigarette smoking, it seems important to mention HIF-1 signaling pathways activated by cigarette smoking that might contribute in several mechanisms to aggravation and development of OSA, among others, through increased BMI (because of metabolic disturbance) – doi: 10.1038/srep34424, 10.3389/fphys.2020.01035, 10.1111/jsr.12995.

-       The effect of nicotine on pain sensitivity and further arousal threshold can also be mentioned in the manuscript – doi: 10.1097/j.pain.0000000000001874, 10.3390/ijms23169080

-       It would enhance the manuscript greatly to add a graphic summarizing the relationship between OSA and cigarette smoking, as well as a table summarizing the effect of disused drugs on the relationship.

Author Response

We would like to thank the reviewer for all the comments in order to improve our manuscript.

The undertaken manuscript widely describes the relationship between cigarette smoking and OSA, which is an important topic in clinical practice, possibly with too little focus in the present research. 

Several notes:

-       While describing possible mechanisms connecting the OSA and cigarette smoking, it seems important to mention HIF-1 signaling pathways activated by cigarette smoking that might contribute in several mechanisms to aggravation and development of OSA, among others, through increased BMI (because of metabolic disturbance) – doi: 10.1038/srep34424, 10.3389/fphys.2020.01035, 10.1111/jsr.12995.

Answer:

Thank you for your comment we have added this mechanism in the revised manuscript.

      Cigarette smoke has been found to reversibly activate hypoxia-inducible factor 1 (HIF-1α) [Daijo, H., Hoshino, Y., Kai, S., Suzuki, K., Nishi, K., Matsuo, Y., Harada, H., & Hirota, K. (2016). Cigarette smoke reversibly activates hypoxia-inducible factor 1 in a reactive oxygen species-dependent manner. Scientific reports6, 34424. https://doi.org/10.1038/srep34424] and this factor has been also found unregulated in OSA patients.  As HIF-1α  has been  involved in the  regulation of metabolic processes and in the  development of  insulin resistance and diabetes , it may further contribute to the development and aggravation of the metabolic co-morbidities of OSA [Gabryelska, A., Karuga, F. F., Szmyd, B., & BiaÅ‚asiewicz, P. (2020). HIF-1α as a Mediator of Insulin Resistance, T2DM, and Its Complications: Potential Links With Obstructive Sleep Apnea. Frontiers in physiology11, 1035. https://doi.org/10.3389/fphys.2020.01035]

-       The effect of nicotine on pain sensitivity and further arousal threshold can also be mentioned in the manuscript – doi: 10.1097/j.pain.0000000000001874, 10.3390/ijms23169080

Answer:

Thank you for your comment we have added this mechanism in the revised manuscript

Sleep fragmentation and sleep loss, which are the main consequences of OSA, were found to enhance pain sensitivity especially through inflammatory and hypoxia markers, as HIF 1a [LaRowe, L. R., & Ditre, J. W. (2020). Pain, nicotine, and tobacco smoking: current state of the science. Pain, 161(8), 1688–1693. https://doi.org/10.1097/j.pain.0000000000001874 ]. Smoking has been related to dysregulated pain processing and the development of persistent pain. On the other hand, pain has been reported to be a barrier to smoking cessation as smokers with pain present more severe withdrawal symptoms and experience more difficulties during quitting attempts with more frequent relapses.[   Kaczmarski, P.; Karuga, F.F.; Szmyd, B.; Sochal, M.; BiaÅ‚asiewicz, P.; Strzelecki, D.; Gabryelska, A. The Role of Inflammation, Hypoxia, and Opioid Receptor Expression in Pain Modulation in Patients Suffering from Obstructive Sleep Apnea. Int. J. Mol. Sci. 202223, 9080. ]

-       It would enhance the manuscript greatly to add a graphic summarizing the relationship between OSA and cigarette smoking, as well as a table summarizing the effect of disused drugs on the relationship.

Answer:

Thank you for your comment we have added the Table and Figure that you have suggested in the revised manuscript.

Round 2

Reviewer 2 Report

The authors answered all the comments to my satisfaction.